

# 1  Aerosol radiative effects with dual view AOD retrievals

Stefan Kinne[1], Peter North[2], Kevin Pearson[2], Thomas Popp[3]
[1]MPI-Meteorology, Hamburg, Germany
[2]Swansea University, Swansea, Great Britain
[3]DLR, Pberpfaffenhofen, Germany
*Correspondence to*: Stefan Kinne (Stefan.Kinne@mpimet.mpg.de)
**Abstract.** Seasonal maps of dual view retrieved mid-visible AOD and AODf for four selected years (1998, 2008, 2019, 2020) are
introduced and assessed in comparisons to MODIS retrievals and general data of an aerosol climatology. Due to different sensor
capabilities (ATSR-2, AATSR and SLSTR) there are still unresolved inconsistencies so that decadal regional trends are not as
detectable as with MODIS retrievals. SLSTR retrieval, however, agree with MODIS retrievals that 2020 Covid impacts on AOD
values (via comparisons to the pre-COVID 2019 reference) are at best minor and secondary to natural anomalies by wildfires and
dust. In radiative transfer applications the dual view AOD data for the four years are processed in the MAC climatology
environment to determine aerosol associated radiative effects for total aerosol and for anthropogenic aerosol. Even though the
calculated radiative effects are affected by retrieval AOD retrieval tendencies, climate relevant TOA net-flux changes are
consistent to result with AOD data from other satellite sensors and a general climatology: -0.9W/m2 for total aerosol with a
significant greenhouse effect and -0.8 and -0.2W/m2 for anthropogenic aerosol with and without indirect effects, respectively.
Aside from global averages, seasonal maps highlight the diversity of regional and seasonal radiative effects.

## 19  1 Introduction

One decade ago, ESA's Aerosol Climate Change Initiative (CCI) set out to develop and improve retrievals
for European satellite sensors. New products for aerosol properties were developed and user case studies were
encouraged to demonstrate dataset usefulness for various applications, as part of the Aerosol_cci+ effort.
One such study here investigates aerosol associated radiative effects, by applying ESA's most advanced
global aerosol datasets, which are based on the University of Swansea dual view retrievals for ATSR-2 [1997-2003],
AATSR [2002-2012] and SLSTR [2017-2020]) satellite sensors with multi-spectral and dual (nadir and forward /
rearward) viewing capabilities  (*North et al. 2021, North 1999*). Four example years (1998, 2008, 2019 and 2020)
were picked to address (1) temporal change by comparing 1998, 2008 and 2019 data and (2) potential COVID
behavior impacts by comparing 2020 with 2019 data.

In the first part, seasonal averages of monthly gridded 1°x1° (latitude/longitude) level 3 aerosol optical

depth (AOD) products for the four years are compared to same year MODIS sensor AOD retrievals (*Levy et al. 2013,
Ginoux et al., 2012*) and to AOD data of the MAC aerosol climatology (*Kinne 2019*). MAC data are not associated
with any specific year, but MAC captures general AOD expected patterns. In MAC, median AOD values from global
aerosol component modeling with current emissions were forced to more realistic conditions by applying regional
adjustments based on multi-annual monthly ground-based network AOD monitoring statistics (Aeronet). For a better
understanding of maxima, decadal changes and differences are examined not only for total AOD but also fine-mode
AOD (AODf). Hereby, AODf is the AOD contribution by (smaller) sub-micrometer size aerosol particles. These
smaller sizes originate mainly from wildfires and pollution (thus also from anthropogenic sources) and by
dominating aerosol number concentrations are particularly important for aerosol impacts on clouds.





In the second part, ATSR-2, AATSR and SLSTR monthly maps for AOD and AODf of all four years (1998,
2008, 2019 and 2020) are applied in an off-line radiative transfer environment to determine aerosol associated
radiative effects. In addition to effects by all aerosols also estimates for today's anthropogenic aerosol are provided –
without and with considering (first) indirect effects via aerosol modified water clouds. To illustrate spatial and
temporal variability impacts are presented in seasonal maps for radiative net flux changes (1) at the top of the
atmosphere (TOA), (2) at the surface and by their difference (3) for the atmosphere. Net flux changes at TOA
capture the climate impact (with a cooling indicated by negative changes and a warming indicated by positive
changes). Net flux reductions at the surface are relevant for surface exchange processes. And solar aerosol layer
heating of atmospheric impacts affect atmospheric dynamics (e.g. lifting/stabilizing aerosol layers).
**2 Dual view (DV) radiometer data**
The applied DV data are based on dual-view and multi-spectral SR heritage sensor retrievals by the
University of Swansea (*North et al., 2021*, *North 1999*). Gridded (1°x1° longitude/latitude) monthly level 3 data for
atmospheric aerosol column loads are examined for four different years: 1998 (ATSR-2), 2008 (AATSR), 2019
(SLSTR) and 2020 (SLSTR). The investigated aerosol properties are (1) the aerosol optical depth at 550nm (AOD)
and (2) the fine-mode aerosol optical depth at 550nm (AODf). AODf captures AOD contributions by sub-
micrometer size aerosols only.
For all four years (1998, 2008, 2019 and 2020) seasonal distribution maps and associated global averages
are compared for AOD and AODf in Figure 1.


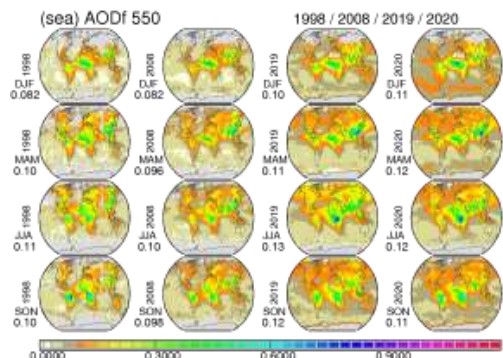

*Figure 1. Seasonal global distribution maps of retrievals for AOD (left panel) and AODf (right panel) with ATSR-2*
*1998 (col1), AATSR 2008 (col2), SLSTR 2019 (col3) and SLSTR 2020 (col4) data. Values to the lower left indicate*
*global seasonal averages.*

ATSR-2 and AATSR sensors have narrower swath widths compared to SLSTR sensors. In addition, the
SLSTR data presented here combine samples of two sensors on two different platforms. To compensate for poorer
statistics by ATSR-2 and AATSR retrievals, comparisons in Figure 1 are presented by season. All DV retrievals
generally agree on the major seasonal maxima. The maxima are related to wildfires over western Africa during DJF,
to dust over Sahara and Gobi deserts and pollution over SE Asia during MAM, to dust over (and off) the Sahara
desert, wildfires over central Africa and pollution over NH urban regions during JJA and to wildfires over southern





America and southern Africa during SON. We identify as remaining inconsistency that the annual global average
AOD and AODf values are about 15% larger for SLSTR compared to ATSR-2 and AATSR. While for all DV-
retrievals the fine-mode AOD fraction (AODf/AOD) for all is (relatively high at) near 65%, there are also consistent
differences with respect to the strength of maxima and of background. SLSTR retrievals yield larger AOD over
oceanic background and strongest wildfire maxima over the Congo during JJA, while ATSR-2 retrievals have
stronger dust maxima over the Sahara during dust seasons (MAM, JJA).

Regional and seasonal AOD and AODf differences are better visualized as differences to the same MAC

climatology (*Kinne, 2019a*) reference in Figure 2.

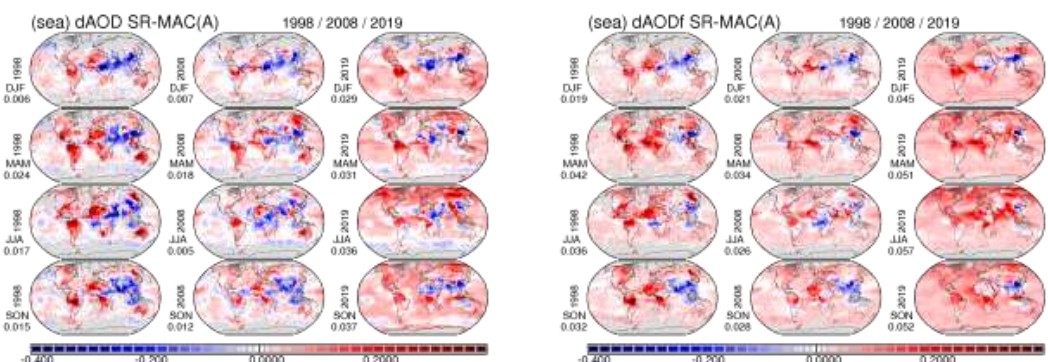

*Figure 2. Seasonal difference maps between DV retrievals for 1998 (col1), 2008 (col2) and 2019 (col3) and the*
*MAC climatology for AOD (left panel) and AODf (right panel). Values to the lower left indicate global seasonal*
*differences.*

Global averages of seasonal AOD are larger than for MAC - by about 15% for ATSR-2 and AATSR and by

about 30% for SLSTR. In addition, there are large regional differences, which vary in strength and even sometimes
in sign with DV sensor retrievals, which are in part caused by different sideways viewing strength and limitations
(SLSTR: forward, ATSR-2 and AATSR: rearward). Thus, attempts to derive decadal trends from DV retrievals are
not yet possible until DV sensor retrieval biases are better understood and corrected.

Deviations of concern to MAC and global modeling are the relatively high AODf attributions (also for dust

outflow across the Atlantic), higher wildfire AOD and lower AOD near the tropics from Africa to Asia in all DV
retrievals. Sensor specific larger deviations to MAC are for SLSTR retrievals the higher AOD background and
higher Northern hemisphere AOD (especially during JJA) and for ATSR-2 the higher AOD over the Sahara with
unexpected high fine mode contributions in ATSR-2.

In a different assessment, MODIS 6.1 Terra retrievals (*Levy et al., 2016 - over oceans, Ginoux et al, 2012 -*

*over land*) seasonal AOD and AODf data are compared for same available years (2008, 2019 and 2020) and
differences are presented in Figure 3.

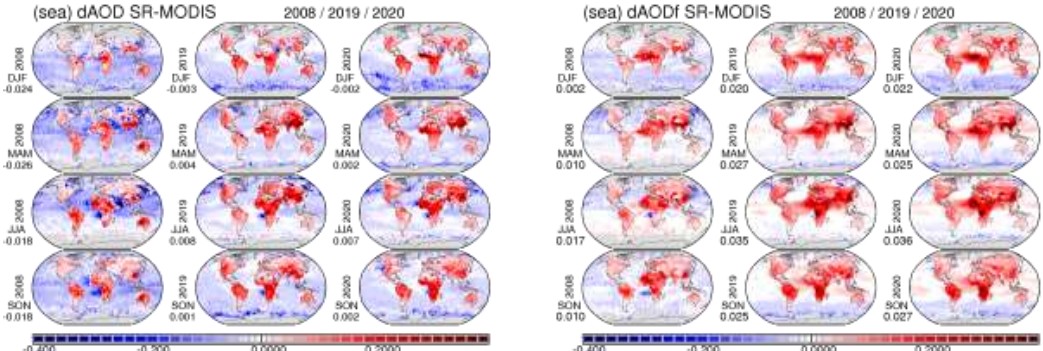

**Figure 3**: *Seasonal difference maps between DV retrievals and MODIS retrievals for 2008 (col1), 2019 (col2) and*
*2020 (col3) for AOD (left panel) and AODf (right panel). Values to the lower left indicate global seasonal differences.*

A first glance at Figure 3 reveals much lower DV AOD data over oceans and much higher AOD and AODf

values over continents - in comparison to MODIS. While larger oceanic AOD by MODIS are likely in error due to
contaminations by clouds (*Hyer et al. 2011*), the larger AOD values over land and the high fine mode AOD fraction
of 65% (vs. about 50% in MODIS and MAC) are of concern in DV retrievals (note, that in DV and MODIS retrieval
methods and assumptions differ over land and ocean).
**3 AOD trends**

DV retrievals from different sensors for selected years are not consistent enough (at this time) to address

decadal trends. Twenty years (2001-2020) of MODIS retrievals (from the same sensor onboard one platform over the
whole period), however, can offer information on recent trends for AODc (=AOD-AODf) and AODf. For
information on semi-decadal trends monthly 1°x1° latitude/longitude AODc and AODf data of MODIS were binned
into five year averages by season. Differences of the three 2006-2010, 2011-2015 and 2016-2020 averages to those
of the 2001-2005 period are presented in Figure 4 (red colors indicate increases and blue colors decreases over time).

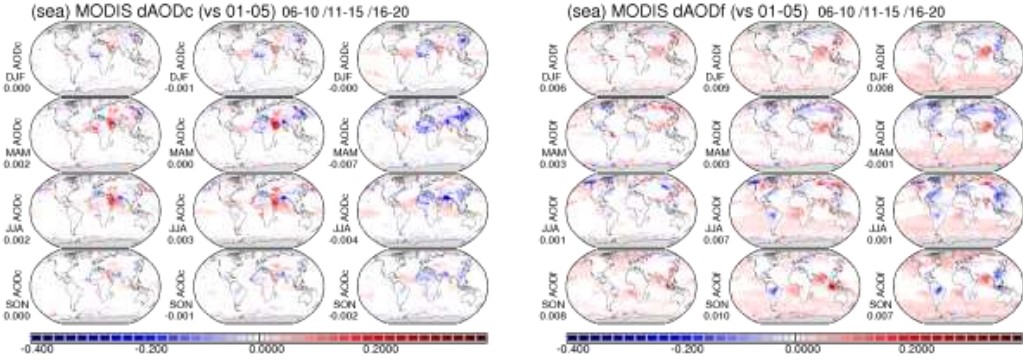

**Figure 4**: *Seasonal maps of semi-decadal trends for AODc (left panel) and AODf (right panel) based on MODIS*
*retrievals. Lower left values indicate global seasonal trends.*





Semi-decadal changes for the total AOD and its AODc and AODf contributions since the year 2000 are
minor in terms of global averages. However, there are regional trends of both signs. For AODc, which mainly
reflects natural variability by mineral dust, there was more dust over Arabia between 2005-2015 (during MAM and
JJA), but less dust over northern India and eastern Asia (during MAM and JJA) and less dust over the Sahara since
2015. For AODf, which represents wildfire activities and pollution strength, less wildfires over southern America
since 2005 (during JJA and SON) but more wildfires over Siberia since 2010 (during JJA) occurred. Over urban
regions there are continuing pollution increases over India (outside the monsoon season), a strong reduction in
pollution though over eastern Asia since 2015 are continued reductions in pollution over the US and EU (during
MAM and JJA).

## 4 COVID-19 impacts

Same sensor SLSTR and MODIS retrievals for the years 2019 and 2020 were compared to explore potential
COVID impacts. Bi-monthly 2020 anomalies with respect to 2019 data for AOD, AODf and AODc differences are
presented in Figure 5.

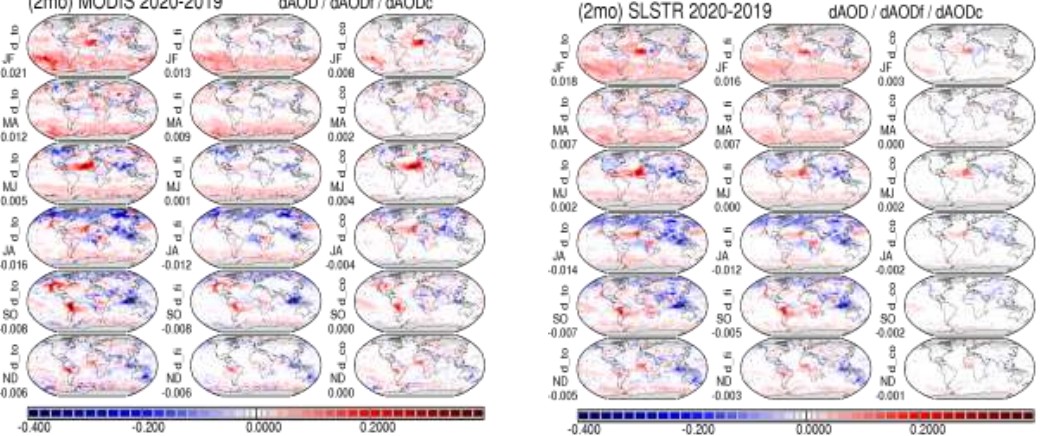

***Figure 5****: Bi-monthly differences between 2020 and 2019 retrievals of MODIS (left panel) and SLSTR (right panel) for AOD (col1), AODf (col2) and AODc (col3). Values to the lower left indicate 2-monthly global 2020-2019 anomaly averages.*

During Covid-19 related shutdown periods (e.g. February and March 2020 over Eastern Asia, May and
April plus October to December 2020 over the U.S. and Europe), reduced traffic may have caused reductions in
AODf (and also in total AOD). However, when examining 2020-2019 AOD differences of both MODIS and SLSTR
in Figure 5, Covid-19 related reduced AOD signatures are difficult to find. Instead, for the 2-monthly (JF, MA, MJ,
JA, SO, ND) regional differences, specific (natural) events are the main modulators for differences.
-      Australian fires early in 2020 yielded positive AODf (and total AOD) anomalies over Southern

Hemispheric higher latitudes during JF and even for MA, as wildfire aerosol was partially convected (and

remained) in the stratosphere.



- Boreal summer wildfires were stronger in 2019 for negative AODf (and total AOD) anomalies over
northern Canada during MJ and JA and over Siberia during JA.
- Indonesian wildfires in the fall of 2019 caused negative AODf (and total AOD) anomalies over Indonesia
during JA and SO.
- Wildfires over the Amazonas region were more intense during 2020 compared to 2019 and yielded positive
AODf (and total AOD) anomalies over Brazil during SO.
- dust transport from the Sahara across the Atlantic were stronger in 2020 than in 2019 and caused positive
AODc (and total AOD) anomalies during MJ and JA.
In summary, Covid-19 impacts on atmospheric aerosol loads and aerosol associated climate impacts are at best
secondary - even on regional scales.
**6 Radiative transfer**
Simulations of atmospheric radiative transfer apply a two-stream radiative transfer model (*Kinne, 2019b*). In
that model, the spectral variability of atmospheric particle properties is represented by separate simulations in eight
solar and twelve infrared spectral sub-bands. In those 20 spectral bands, a total of 120 exponential terms are used to
represent atmospheric trace gas absorption. The daytime solar impact variability is covered by simulations at 9
different solar elevations. Atmospheric state and trace gas properties apply latitude and season relevant AFGL
(*Andersen et al., 1986*) standard atmospheres and the vertical variability is approximated by 20 plane-parallel
homogenous atmospheric layers. Land surface (visible and near-IR) albedo data of MODIS (*Schaaf et al., 2002*) are
applied, over open oceans a sun-elevation dependent albedo is considered (ref) and snow and ice coverage is based
on NOAA microwave data (*Taylor et al., 1996*). Cloud cover and cloud optical depth are represented by multi-
annual monthly ISCCP data (*Rossow et al., 1993*) and eight independent simulations cover all eight possible cloud
(high, mid, low) layer combinations. Simulations at each (of the 64800 1°x1° latitude/longitude) grid locations
averages for each month.
AOD and AODf monthly maps of ATSR-2, AATSR and SLSTR replace those of the MAC aerosol
climatology (*Kinne, 2019a*). In the MAC climatology a monthly median AeroCom modeling background is
regionally and seasonally modified towards available monthly statistics by sun-/sky photometry of AERONET
(*Holben et al., 1999*) and MAN (*Smirnov et al, 2009*). Global modeling provides the needed information on aerosol
vertical distribution and on today's anthropogenic AODf fraction. For aerosol indirect effects by anthropogenic
aerosol a simple relationship from satellite retrievals over oceans linking cloud droplet number concentrations
(CDNC) and AODf is applied. This relationship translates anthropogenic AODf increases in reference to a (e.g. pre-
industrial) AODf background into CDNC increases. Associated decreases to the cloud droplet size (more but smaller
drops - assuming no changes to the cloud water content) define the (anthropogenic) aerosol associated aerosol
indirect effects.
Radiative impacts in the atmosphere are defined by differences of two simulations between a modified and
a standard configuration. All simulations consider ISCCP tropospheric clouds. These 'dual-call' radiative transfer
applications investigate



-    ***impacts of total aerosol presence** (along with shortwave and longwave contributions)*
-    ***impacts of anthropogenic aerosol presence** (at TOA referred to as 'direct forcing')*
-    ***impacts of reduced droplet radii by anthropogenic aerosol** (at TOA: indirect forcing')*

Radiative forcing climate feedbacks (which cannot be considered in dual call simulations) are and can be
ignored, because long-term Earth System Model (ESM) simulations with fixed sea surface temperatures show that
atmospheric feedbacks may introduce forcing deviations of less than 10% (even smaller than AOD uncertainties for
total and especially pre-industrial AOD).
Anthropogenic aerosols are defined via a fraction of smaller sub-micrometer aerosol sizes (from global
modeling). Anthropogenic (land-use) contributions of dust are ignored, because (smaller size) dust radiative effects
are close to neutral. Also as only impacts of aerosols on lower altitude water clouds are allowed, only solar radiative
effects need to be considered for the impacts by anthropogenic aerosol.
**7 global radiative effects**
Aerosol radiative effects are determined separately for all four DV. Global averages of net fluxes at TOA
(negative: climate cooling, positive: climate warming) and at the surface (negative: loss, positive: gain) are summarized
in Table 1.
***Table 1** Global annual averages for today's aerosol radiative effects at TOA and surface*

| | | TOA | | | | | surface | | | | |
|---|---|---|---|---|---|---|---|---|---|---|---|
| | $W/m^2$ | ATSR2 | AATSR | SLSTR | SLSTR | *MAC* | ATSR2 | AATSR | SLSTR | SLSTR | *MAC* |
| | | 1998 | 2008 | 2019 | 2020 | | 1998 | 2008 | 2019 | 2020 | |
| total | sw+lw | - .87 | - .82 | - .95 | - .99 | *- .93* | - 5.8 | - 5.6 | - 6.9 | - 6.9 | *- 4.6* |
| total | sw | - 1.5 | - 1.4 | - 1.5 | - 1.5 | *- 1.7* | - 7.3 | - 7.1 | - 8.4 | - 8.4 | *- 6.4* |
| total | lw | + .66 | + .65 | + .60 | + .59 | *+ .76* | + 1.5 | + 1.5 | + 1.4 | + 1.4 | *+ 1.8* |
| ant | direct | - .24 | - .24 | - .23 | - .22 | *- .23* | - 2.1 | - 2.0 | - 2.6 | - 2.5 | *- 1.6* |
| ant | dir/ind | - .86 | - .86 | - .81 | - .80 | *- .89* | - 2.7 | - 2.7 | - 3.2 | - 3.1 | *- 2.2* |


At TOA, today's total aerosol cools the climate by about -0.9W/m2. Hereby the stronger shortwave cooling
of about -1.5 W/m$^2$ is partially offset by about +0.6 W/m$^2$ of elevated (larger size) mineral dust. For anthropogenic
aerosol impacts, where smaller aerosol sizes contribute, mainly shortwave effects matter. The anthropogenic aerosol
via its added presence causes a climate cooling of about -0.2 W/m$^2$ and when effects by modified water clouds are
added, this climate cooling increases to about -0.8 W/m$^2$.
At the surface, today's aerosol reduces the radiative net flux much stronger than at TOA. Global average
surface net flux reductions of about -5.0 W/m$^2$ are composed of the shortwave losses of about -6.5 W/m$^2$ and
longwave gains of about +1.5 W/m$^2$, mainly due to elevated (larger size) mineral dust. The presence of
anthropogenic (added since pre-industrial times) aerosol reduces surface net fluxes by about -2.4 W/m$^2$, which is
further reduced to about -3.0 W/m$^2$, when impacts by aerosol modified clouds are included.





The radiative effects of aerosol in the atmosphere are captured by the difference of net flux impacts at TOA
and surface. Since reductions of net fluxes are larger at the surface than at TOA, the atmospheric impact is positive
which means an atmospheric heating. Global averages for atmospheric impacts are summarized in Table 2.
***Table 2*** *Global annual averages for today's aerosol solar atmospheric heating*

|  |  |  | **total** |  |  |  |  | **anthropogenic** |  |  |  |
|---|---|---|---|---|---|---|---|---|---|---|---|
|  | *W/m2* | ATSR2 | AATSR | SLSTR | SLSTR | *MAC* |  | ATSR2 | AATSR | SLSTR | SLSTR | *MAC* |
|  |  | 1998 | 2008 | 2019 | 2020 |  |  | 1998 | 2008 | 2019 | 2020 |  |
| atm | sw+lw | + 4.9 | + 4.7 | + 6.0 | + 6.0 | *+ 3.7* |  | + 1.9 | + 1.8 | + 2.4 | + 2.3 | *+ 1.3* |


For the atmosphere, today's total aerosol heats the atmosphere on average by 5 W/m$^2$. Hereby, a 6 W/m$^2$
average solar heating is partially offset by an average 1 W/m$^2$ in infrared cooling. Atmospheric solar heating by
anthropogenic aerosol is about +2 W/m$^2$.
The global averages for the four DV retrieval years are similar at TOA to each other and even to standard
MAC simulations, which apply AOD and AODf maps of the MAC climatology. Since these MAC AOD values are
on average smaller than those of the DV retrievals, especially for SLSTR (as illustrated in Figure 2), SR retrieval
data associated global impacts are on average larger at the surface (and the atmosphere).
**7 other influential quantities**
To better understand aerosol radiative impacts, it is important to explain that aside from data for AOD
(and/or AODf) also other aerosol properties and several environmental properties are influential. Any such influence
would also need to be considered for the radiative forcing estimates, if any of those other quantities experienced a
trend during an observation period for AOD.
Important aerosol properties, other than AOD, are aerosol absorption strength, aerosol size and aerosol
altitude. Sensitivity calculations with the two-stream radiative transfer code show that if aerosol absorption
efficiency (i.e. 1-SSA) increases, then net flux losses at the surface are increased and at the same time TOA impacts
become less negative - both changes increase the inside atmosphere impacts. If aerosol includes contributions of
larger super-micrometer sizes (quantified by AODc), then interactions with longwave radiation cannot be ignored
anymore (in contrast to smaller wildfire and pollution aerosol). Longwave (and greenhouse / warming) effects
increase with aerosol absorption and aerosol elevation. Thus, elevated absorbing super-micrometer mineral dust can
yield significant longwave effects.
Important environmental properties are the incoming solar radiation, the solar elevation, the co-location
with clouds (and even with trace gases) and the surface radiative properties, such as solar albedo and temperature.
Every aerosol solar impact requires sunlight and the aerosol (back-) scattering is strongest near solar elevations of 20
degrees. Very important for TOA impacts is the co-location of aerosol with clouds. With clouds, aerosol (presence)
impacts are at most secondary - except for the case of absorbing aerosol above clouds: When aerosol dims the solar
reflection of clouds (to space), then the solar TOA impact of aerosols reverses to a warming. Similar to the 'aerosol
above cloud' scenario also an increasing brighter surface albedo reduces the potential for aerosols to cool to the point
that for (even weakly) absorbing aerosols over bright surfaces (e.g. snow) an aerosol TOA warming (by dimming)





252 the solar surface albedo can be expected. Surface temperature, temperature of the aerosol layer and trace gas

253 distributions only matter for super-micrometer size aerosols for longwave (greenhouse and re-radiation) effects.

## 8 seasonal and regional radiative effects

255   The global average impacts of Tables 1 and 2 are now brought to life in the subsequent presentation of

256 seasonal impact maps. With the average aerosol lifetime of a few days and with differences in solar insolation,

257 surface albedo and clouds, regional and seasonal impacts are highly diverse - sometimes even with changes to the

258 sign in the impact and with local absolute maxima up to an order of magnitude larger than global average values.

### 8.1 TOA net-flux impact distributions

260   Seasonal aerosol radiative effects at TOA associated with the four DV AOD retrievals for total aerosol

261 (presence) are presented in Figure 6, which also informs on shortwave and longwave contributions via annual

262 averages.

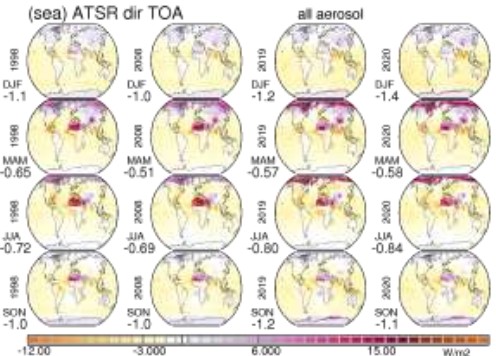
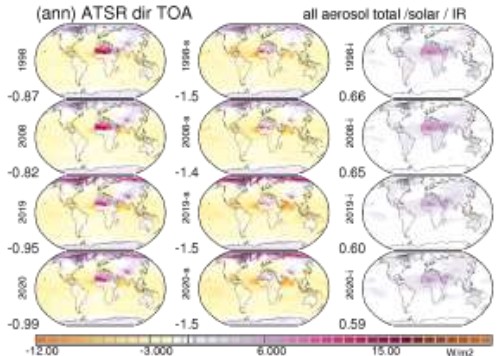

*Figure 6. Simulated total aerosol (presence) radiative effects at TOA with AOD data of ATSR-2 1998, AATSR 2008, SLSTR 2019 and SLSTR 2020. Compared are seasonal impacts (left panel) and annual impacts (right panel), with separations into its shortwave (-s) and longwave (-l) contributions. Positive values represent a warming and negative values a cooling. Values to the lower left indicate global averages.*

271   Aerosol direct TOA radiative effects for total aerosol vary globally between - 0.8 and -1.2 W/m² (with the

272 stronger SLSTR cooling linked to its larger AOD). On a regional basis, TOA aerosol radiative effects vary between -

273 10 and +25 W/m², with the strongest cooling in dust outflow regions over oceans and the strongest warming over

274 (brighter) deserts. The solar effect is a cooling, except over brighter surfaces of deserts and especially snow/ice (of

275 polar regions). For elevated (even weakly) absorbing mineral dust the associated greenhouse effect contributes with a

276 significant warming, which is largely responsible for the peak warming over the Sahara.

277   Seasonal radiative effects by today's anthropogenic aerosol at TOA for the direct (aerosol presence) and the

278 combined direct and indirect effect are presented for the four DV AODf (and AOD) retrievals in Figure 7.






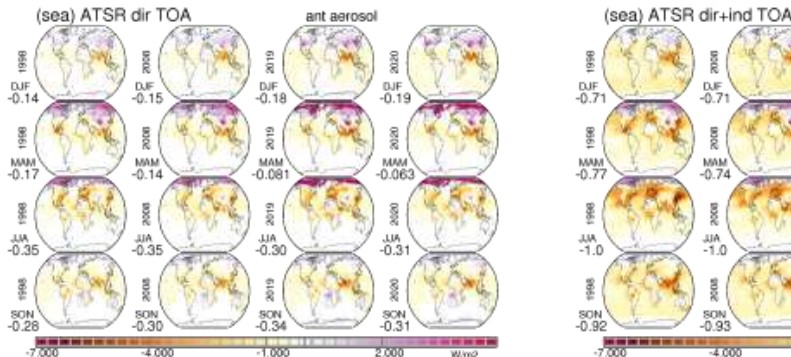

**Figure 7**: *Seasonal maps of simulated anthropogenic aerosol radiative effects at TOA with AODf data of ATSR-2*
*1998 (col1), AATSR 2008 (col2), SLSTR 2019 (col3) and SLSTR 2020 (col4). Positive values represent a warming*
*and negative values a cooling. Direct (aerosol presence) impact maps (left panel) and combined direct and indirect*
*(via modified clouds) impact maps (right panel) are presented. Values to the lower left are global seasonal averages.*
The direct aerosol (presence) forcing (TOA aerosol radiative effects at all-sky conditions) varies between -
0.24 and -0.29 W/m$^2$, with the strongest cooling suggested by SLSTR and their larger AOD. Regionally and seasonally
the aerosol direct forcing varies between -6.0 W/m$^2$ (pollution outflow offshore of urban regions during the summer)
and +4.0 W/m$^2$ (absorbing aerosol over snow). The small warming during the SON season over the SE Atlantic is
caused by the outflow of elevated absorbing wildfire aerosol as it dims the solar albedo of lower stratocumulus cloud
fields. Overall regions with climate cooling dominate.
The combined direct and indirect (considering aerosol induced reductions to the droplet size of water clouds)
aerosol forcing is almost identical at -0.87 W/m$^2$. Regionally and seasonally, the combined anthropogenic forcing
varies between -6.0 and +2.0 W/m$^2$. Compared to direct forcing the climate cooling over urban regions and over (dark)
oceanic regions with pollution outflow is increased. Also, the direct SON dimming over the Southeast Atlantic is gone.
The JJA season contributes with the strongest cooling.
**8.2 surface net-flux impact distributions**
Seasonal aerosol radiative effects at the surface associated with the four DV AOD retrievals for total aerosol
(presence) are presented in Figure 8, which also informs on shortwave and longwave contributions via annual
averages.
Aerosol direct radiative effects for total aerosol at the surface vary globally between -5.4 and -6.4 W/m$^2$
(with the stronger SLSTR cooling linked to its larger AOD). On a regional basis the surface aerosol radiative effects
vary between -40 and 0 W/m$^2$. The strongest cooling occurs near continental aerosol source regions, especially if
large AOD data combine with a strong aerosol absorption potential (of wildfires and urban pollution). Aside from
reductions to the solar radiation there are also IR re-radiation gains by elevated (colder) dust, so that for mineral dust
impacts reductions of solar net fluxes at the desert surface are almost compensated by infrared net flux gains.

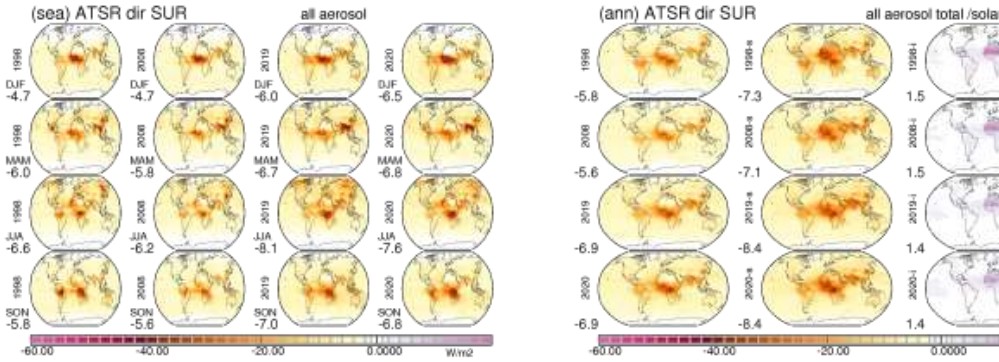

**Figure 8.** *Simulated total aerosol (presence) radiative effects at the surface with AOD data of ATSR-2 1998, AATSR 2008, SLSTR 2019 and SLSTR 2020. Compared are seasonal impacts (left panel) and annual averages (right panel) with separations into its shortwave (-s) and longwave (-l) contributions. Positive values represent surface net-flux gains and negative values surface net-flux losses. Values indicate global averages.*

Seasonal aerosol radiative effects by today's anthropogenic aerosol at the surface for the direct (aerosol presence) and for the combined direct and indirect effect are presented for the four DV AODf retrievals in Figure 9.

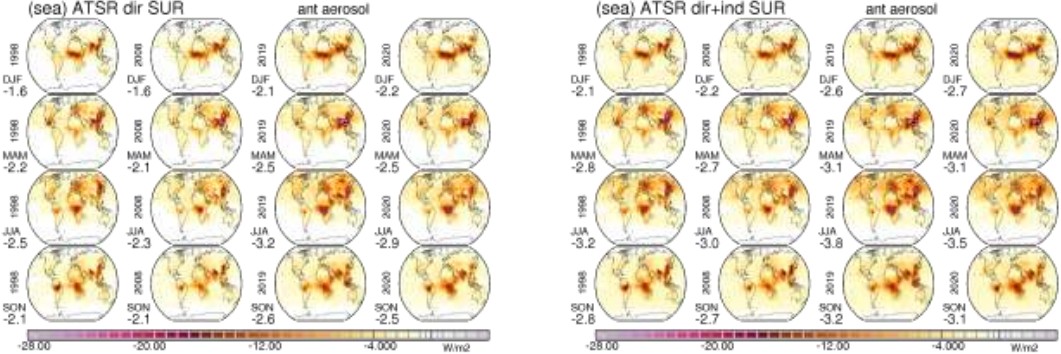

**Figure 9**. *Seasonal maps of simulated anthropogenic aerosol radiative effects at the surface with AODf data of ATSR-2 1998 (col1), AATSR 2008 (col2), SLSTR 2019 (col3) and SLSTR 2020 (col4). There are only negative values which represent net flux losses at the surface. Direct (aerosol presence) impact maps (left panel) and combined direct and indirect (via modified clouds) impact maps (right panel) are presented. Values to the lower left are global seasonal averages.*

The direct radiative effects by anthropogenic aerosol at the surface vary globally between -1.5 and -1.9 W/m², with stronger SLSTR net flux reduction linked to its larger AOD. All surface radiative effects are only negative and can be as large as -20 W/m². The strongest cooling occurs over continental regions and seasons with (strongly absorbing) wildfire and (absorbing) urban pollution. Maximum solar net flux reductions are over Eastern Asia during MAM and for SLSTR data over the Congo regions during JJA.

The combined direct and indirect radiative effects by anthropogenic aerosol at the surface vary globally between -2.6 and -3.1 W/m² with stronger SLSTR net flux reduction linked to its larger AOD. All surface radiative




effects are negative and slightly more negative than the direct effect. Compared to the direct impact, net flux
reductions at the surface are mainly increased over (dark) ocean regions affected by pollution outflow.
**8.2 atmospheric net-flux impact distributions**
Seasonal aerosol radiative effects in the atmosphere for total and anthropogenic aerosol are presented for the
four DV AOD and AODf retrievals in Figure 10.

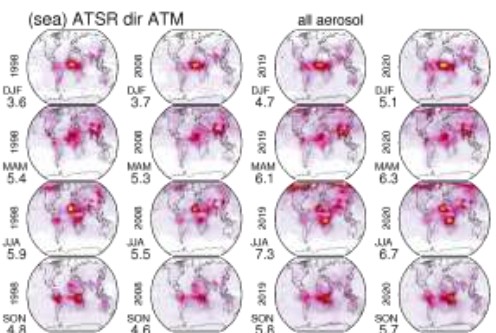
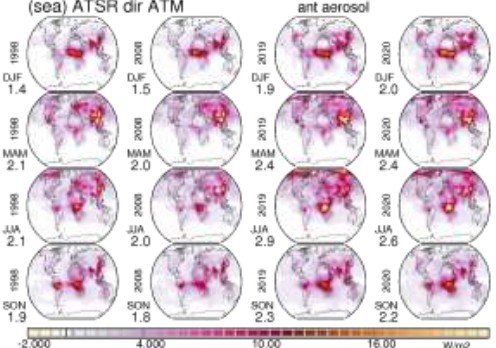

*Figure 10: Seasonal maps of simulated aerosol radiative impacts on the atmosphere with AOD and AODf data of*
*ATSR-2 1998 (col1), AATSR 2008 (col2), SLSTR 2019 (col3) and SLSTR 2020 (col4). Positive values by shortwave*
*heating dominate. Atmospheric impacts are presented for total (left panel) and anthropogenic aerosol (right panel).*
*Values to the lower left indicate global seasonal averages.*

The direct radiative effect in the atmosphere varies globally for total aerosol between +4.5 and +5.7 W/m$^2$
and for anthropogenic aerosol between +1.7 and +2.2 W/m$^2$. The stronger atmospheric heating with SLSTR data is
due to its larger AOD. The aerosol atmospheric effects are positive and can reach above +30 W/m$^2$ over regions with
strong aerosol absorption values. The strongest solar heating is over the biomass burning regions of western Africa in
the DJF season and over the Congo region in the JJA season.
**9 discussions**
The strengths of simulated aerosol radiative impacts are certainly modulated by the aerosol optical depth
maps used, as demonstrated by the larger SLSTR impacts with their larger AOD (compared to ATSR-2, AATSR and
MAC). For more accurate impact estimates more accurate and more consistent satellite data (also to other sensor
retrievals for the same period) in terms of distributions and maxima are required. Another aspect, especially for the
accuracy of anthropogenic impacts, is the AOD attribution to smaller aerosol sizes. The DV retrievals have a
AODf/AOD ratio, which is much larger than in MODIS retrievals or global modeling. In general, higher AODf
values yield stronger anthropogenic direct effects but weaker indirect effects - via a higher pre-industrial reference
background. Despite open issues in DV retrievals for AOD and AODf, the regional and seasonal aerosol radiative
effects maps offer systematic insights and context to (single value) global averages.







**10 summary**

A user case study first examined and then applied (in a radiative transfer study) monthly 1°x1° latitude/longitude gridded mid-visible AOD and AODf data based on dual view (DV) retrievals for four selected years: 1998 with the ATSR-2 sensor, 2008 with the AATSR sensor and 2019 and 2020 with SLSTR sensors from two different platforms. DV-retrieved AOD maps differed mainly due to different sensor capabilities with higher oceanic background and higher NH continental summer values in SLSTR. While any extraction of decadal change by comparing 1998, 2008 and 2019 DV data is discouraged, long term changes for AOD, AODc and AODf between 2001 and 2020 are offered by consistently retrieved MODIS sensor data. Even in comparisons of five year averages, variability of major natural aerosol events dominate. Still, with respect to anthropogenic change, AODf changes over urban regions document (1) continued pollution increase over (northern) India outside the Monsson season, (2) strong reductions over eastern Asia since 2015 and (3) ongoing reductions in pollution over the U.S. and EU (during MAM and JJA). Comparing 2020 with 2019 data, signals of Covid-19 related reductions of AODf (due to reduced industrial activities and traffic) were ay best secondary to natural anomalies both with SLSTR and MODIS data.

In an application, the DV AOD and AODf maps for the four selected years were applied in radiative transfer simulations to determine associated aerosol radiative effects for total aerosol and for today's anthropogenic aerosol. The calculated radiative effects for all four years are similar, with larger effects usually associated with the higher AOD in SLSTR (over oceans and most continental regions) compared to ATSR-2 and AATSR. At TOA, though, global average impacts with AOD and AODf maps for all four years were similar with a -0.9 W/m$^2$ cooling for total aerosol and a -0.8 W/m$^2$ climate cooling for anthropogenic aerosol - including a -0.6 W/m$^2$ cooling by (first) indirect effects through clouds.

For radiative impacts of total aerosol it is also demonstrated that with the presence of elevated mineral dust longwave radiative impacts must be included, hereby offsetting on a global average about 20% of the solar impacts. Seasonal impact maps are offered to detail the highly diverse seasonal and regional nature of aerosol radiative effects, which are often a magnitude larger than their associated global averages and at TOA, radiative effects even can switch their sign.

**10 acknowledgements**

This work was supported by the European Space Agency as part of the Aerosol_cci+ project (ESA Contract No. 4000126239/19/I-NB). The provision of MODIS collection 6.1 aerosol retrievals by Rob Levy (NASA-GSFC) and by Paul Ginoux (NOAA) for comparisons and decadel change is appreciated.





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
