# Peer review of "Aerosol radiative effects with dual view AOD retrievals"

_Atmospheric Chemistry and Physics, 2021_

## Author Comment (AC2)

*Responses* to reviewer 2 comments

The study is essentially four papers in one, often loosely linked to the stated aim of being a user case study of an Aerosol_cci product. Unfortunately none of the four items are analysed with enough depth and novelty to recommend publication.

*These are all valid points and efforts are underway to provide more details – in particular address associated retrieval issues.*

The first item, in section 2, focuses on the performance of the AOD and AODf dual-view retrievals based on "SR heritage" sensors. That analysis is clearly premature. As the authors write on line 89, there is a clear need for "DV sensor retrieval biases [to be] better understood and corrected". The SR retrieval teams should first publish a paper documenting time series of AOD and AODf over 1998-2020 with the different SR sensors, identifying possible global biases and coming up with ways to splice the different datasets consistently.

*This is a good suggestion. It was planned to bring in the new SLSTR data with the lastest processing and compare those with reprocessed ATSR2 and AATSR data. At this time a complete re-processing of the time-series is not available and there is the 2012-2018 DV-type retrieval sensor gap. Extra efforts will be done to address retrieval and sensor differences.*

For the second item, in section 3, the authors give up on dual-view retrievals and use MODIS to very briefly look at decadal trends in aerosol optical depth. That analysis is out of the stated scope of the paper, and does not really add value to previous work, for example
https://doi.org/10.5194/acp-20-139-2020

*The plan here is now to fully include MODIS data (with changes to the title), which are more consistent and cover continuously from 2001 to 2020 … also for subsequent radiative forcing impacts*

The third item, in section 4, looks at possible changes in AOD from decreases in activity due to the Covid pandemic. That is an interesting question, but which requires a more careful analysis than done here. First, many aerosol sources remained active during lockdowns (https://doi.org/10.1029/2020GL088913) so the impact of AOD is smaller than could be anticipated. In addition, one cannot simply compare 2020 to 2019, because that ignores interannual variability (https://doi.org/10.1029/2020GL091805, https://doi.org/10.1029/2021GL093841, https://doi.org/10.1029/2020gl091699), the complexity of aerosol chemistry (https://doi.org/10.1029/2020GL088533, https://doi.org/10.1093/nsr/nwaa137), and a potential masking of the Covid signal by anomalous meteorology in 2020 (https://doi.org/10.1029/2020JD034090, https://doi.org/10.1126/science.abb7431).

*The study applies 1x1 deg monthly average data. This limits any more detailed study. The point here was to show that COVID likely impacts on regional atmospheric aerosol loads (e.g. AOD or AODf) are small (to negligible) to aerosol load changes by natural anomalies of wildfires,*

*even if some COVID signals are modulated by meteorology and chemistry (as investigated in the given references).*

The fourth item, in sections 6-8, looks at the impact of using DV-retrieved AOD fields on aerosol radiative effects in the MAC climatology framework. Differences can be explained by differences in AOD, and the authors find that assumptions on the single-scattering albedo have a large impact. Here, the use of DV retrievals feels unnecessary to draw these conclusions. The same analysis could have been done within MAC itself, and indeed most conclusions can already be drawn from the analysis by Kinne et al. 2019 (https://doi.org/10.5194/acp-19-10919-2019).

*The forcing differences will be examined in more detail now including results with MODIS data and comparisons to data from global modeling. Aside from AOD fields also other environmental properties are influential and in order to examine the absorption impacts (in separate rad. effect simulations) also the DV retrieval associated AAOD (in place of the MAC) will be prescribed. The presentation focus here is on seasonal maps.*

Other comments:

Section 2: The presentation of the three SR sensors in that section is minimal, yet the importance of their different capabilities between them and with MODIS is invoked very early in the analysis. What are the different swaths? What are the different fractional coverages?

*More details will be provided from the retrievals groups*

Lines 16-17: "aerosol with a significant greenhouse effect" Need to clarify that this expression refers to longwave radiative effects.

*Ok*

Lines 31-32: "not associated with any specific year". That cannot be correct, since the reader is told a few lines later that MAC is suitable for looking at decadal trends.

*MAC is NOT associated with a specific year and NOT used for decadal trends*

Lines 53-54: Does "size" refer to diameter or radius? Is AODf defined in the same way in the DV dataset, in MODIS, and in MAC?

*Size is split at radii of 0.5um in MAC and ATSR. There may be slight changes in the MODIS definitions over oceans (Levy) and land (Ginoux). These differences will be addressed in an update but split-size differences a secondary compared to the huge differences in the fine-mode fractions between DV and MAC of MODIS.*

Lines 87-88: Why would the type of sideway viewing influence seasonal statistics?

*The DV retrieval group will provide more info in an updated version.*

Technical comments:

- Many acronyms are never spelled out.

*will be fixed*

- There is no section 5 but there are two sections 7.

*thanks*